# Mix and Reason: Reasoning over Semantic Topology with Data Mixing for Domain Generalization

**Chaoqi Chen**[1] **Luyao Tang**[2] **Feng Liu**[3] **Gangming Zhao**[1] **Yue Huang**[2] **Yizhou Yu**[1*]

[1]The University of Hong Kong    [2]Xiamen University    [3]Deepwise AI Lab

cqchen1994@gmail.com, lytang@stu.xmu.edu.cn, liufeng@deepwise.com
gmzhao@connect.hku.hk, yhuang2010@xmu.edu.cn, yizhouy@acm.org

## Abstract

Domain generalization (DG) enables generalizing a learning machine from multiple seen source domains to an unseen target one. The general objective of DG methods is to learn semantic representations that are independent of domain labels, which is theoretically sound but empirically challenged due to the complex mixture of common and domain-specific factors. Although disentangling the representations into two disjoint parts has been gaining momentum in DG, the strong presumption over the data limits its efficacy in many real-world scenarios. In this paper, we propose Mix and Reason (`MiRe`), a new DG framework that learns semantic representations via enforcing the structural invariance of semantic topology. `MiRe` consists of two key components, namely, Category-aware Data Mixing (CDM) and Adaptive Semantic Topology Refinement (ASTR). CDM mixes two images from different domains in virtue of activation maps generated by two complementary classification losses, making the classifier focus on the representations of semantic objects. ASTR introduces relation graphs to represent semantic topology, which is progressively refined via the interactions between local feature aggregation and global cross-domain relational reasoning. Experiments on multiple DG benchmarks validate the effectiveness and robustness of the proposed `MiRe`.

## 1 Introduction

In real-world scenarios, such as autonomous driving and computer-aided diagnosis, deep neural networks are expected to be trained on data collected from a small number of domains, but with the objective of being deployed across novel domains with different data distributions to reduce annotation costs and meet a broader range of needs. However, out-of-distribution data [29] does not satisfy the basic IID assumption of modern deep learning models and emerges as an inevitable challenge, significantly hindering the deployment of source-trained models in new target domains. Domain generalization (DG) [47, 6, 23], which addresses this challenge by learning invariant representations across multiple source domains, has attract a great deal of attention in the research community.

The objective of DG is to recover latent semantic factors that are independent of domain. Most DG approaches rely on the assumption that the latent representations can be divided into two disjoint components, *i.e.,* common and domain-specific components. Inspired by this, mainstream DG paradigms include data/feature augmentation [68, 75], feature disentanglement [53, 40], invariant risk minimization [3, 2], meta-learning [33, 17], to name a few. By facilitating common knowledge extraction and alleviating domain-specific components, these approaches have made tremendous progress in various tasks, such as image classification [6, 84] and semantic segmentation [37, 51].

---

*Corresponding author

36th Conference on Neural Information Processing Systems (NeurIPS 2022).

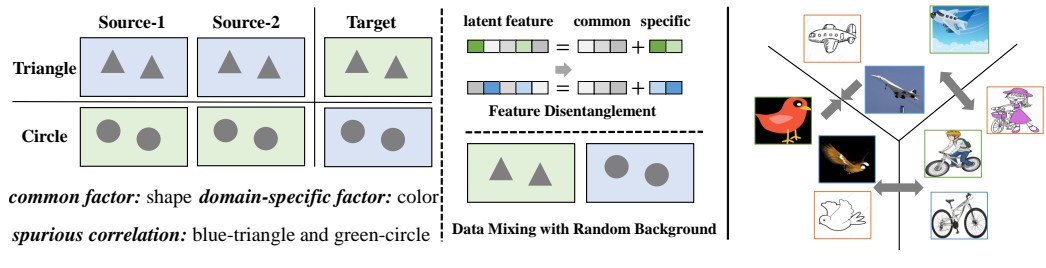

**(a)** Ideal Case  **(b)** Real-World Case

Figure 1: **(a)** In ideal case, we assume that the data has two attributes (shape and color). **(b)** In real-world case, we observe that bird and airplane share more similarity as the design of airplane is motivated by bird, while both of them are dissimilar with bicycle. Thus, the embedding of bird and airplane should be closer and their connecting weight should be enlarged, which are independent of domain and robust to unseen test environments. We call this property as *structural invariance*.

Despite the promise, existing DG approaches may still be confined by two critical limitations. (1) Traditional DG methods [47, 36, 6, 34, 17] strive to enforce invariance between domains but ignore the potential spurious correlations between common and specific components. Take Fig. 1 (a) as an example, the correlation between shape and color may induce the classifier to use background colors as the classification evidence. (2) To statistically avoid the spurious correlations, a body of research [54, 53, 16, 44, 61, 40] proposes to identify the common component through latent factor disentanglement. In this regard, causal graphs [50] provide promising theoretical guarantees. As shown in Fig. 1 (a), previous disentanglement-based methods manifest remarkable success in simulated data, requiring that models have seen some distribution of values for an attribute [71] (*e.g.,* shape and color in DSPRITES [46]), which is too restrictive to be satisfied. In real-world case (cf. Fig. 1 (b)), we cannot (*i*) explicitly know the latent feature is composed of how many factors, (*ii*) identify which factors are semantic-related or not, and (*iii*) how these potential factors interact. Since we have no a *priori* knowledge or assumptions about the attributes (factors), it is prohibitively difficult to disentangle them in an unsupervised manner [42], largely impeding the practical use of these approaches. On the other hand, the internal relations among different classes are neglected, which are crucial for the robustness of learned models to distribution shifts. This finding draws motivation from the human intelligence that when learning new concepts, humans are talented at *comparing* and *reasoning* [26, 15, 5]. In that sense, we aim to endow the classifier with the ability of perceiving and maintaining structural semantic relations rather than ideally seeking *clean* semantic[1] representations.

Grounded on the above discussions, we propose Mix and Reason (`MiRe`), a simple yet effective DG approach, which enables generalization via data-level augmentation and feature-level regularization. The proposed `MiRe` consists of two key components, Category-aware Data Mixing (CDM) and Adaptive Semantic Topology Refinement (ASTR), which learn semantic representations by enforcing structural invariance of semantic topology across domains in the embedding space. CDM induces the classifier to focus on recognizing foreground objects as their supporting evidence for classification, alleviating the potential influence of domain-specific factors (*e.g.,* scene layout and style). Specifically, foreground region is derived via integrating two complementary network activation maps generated by category and domain classification losses, while background is randomly cropped from images sampled from other domains with Gaussian blur. ASTR introduces the concept of semantic topology, which is instantiated as the relation graphs, to progressively refine the representations (node) and the topological relations (edge) of semantic anchors. Semantic anchor is denoted by the centroids of category-wise features in per domain, largely reducing the intra-domain variations. To be specific, each input instance is compared to the set of semantic anchors and dynamically aggregate features according to the global[2] semantic topology and local adjacent relations. Then, the global features are updated by the aggregated local features. Finally, a bipartite graph network is developed on the top of semantic topology to perform cross-domain relational reasoning and induce structural invariance.

---

[1]The word 'semantic' and 'common' are used interchangeably, but 'invariance' does not equal to them. Invariance may contain spurious correlation, which can be alleviated but is difficult to be absolutely eliminated.

[2]The word 'global' denotes that the features are computed from the whole domain (dataset), and the word 'local' denotes that the features are computed from mini-batch samples at each iterations.

Experiments on four standard DG benchmarks (PACS, VLCS, Office-Home, and DomainNet) reveal that `MiRe` exceeds the performance of state-of-the-art methods. In addition, CDM is applied as a plug-and-play module for state-of-the-art DG methods (such as MixStyle [84] and EFDM [78]) to boost their performance. The robustness of ASTR is further evaluated on a challenging Chest X-ray benchmark [44] that has explicit spurious correlation.

## 2 Related Work

**Domain Generalization.**    Generalizing a well-trained model to novel environments with different data distributions is a challenging and long-standing machine learning problem [81, 69]. Current state-of-the-art DG approaches can be roughly categorized into four types. (1) *Feature Alignment.* Bridging the domain gap through statistic matching and adversarial learning [38, 36, 45] provides invariant representations, which are expected to be shared by novel domains. A major defect is that the learned representations is prone to mix common and domain-specific components, potentially resulting in strong bias towards spurious relations. (2) *Feature Disentanglement.* To address the above issue, disentangling latent feature into two disjoint parts has attract a great surge of interest. Prevailing approaches [54, 53, 16, 44, 61, 40] resort to causal graphs [50] to explicitly identify causal and non-causal factors with theoretical guarantees. If this could be perfectly achieved, learned models will generalize well under any circumstances. However, these theoretical results require strong assumptions, *e.g.,* the degree of diversity between causal and non-causal factors and the presence of an estimated causal graph [62], or a priori knowledge about the combinations of latent factors, *e.g.,* distribution of values for a certain attribute. The complex combinations of many real-world cases (*e.g.,* DomainNet [52]) greatly impede the practical uses. In that sense, a principled disentanglement solution is hard to be reached. (3) *Data/Feature Augmentation.* A simple yet effective approach is to augment the diversity of data or feature so as to avoid overfitting to training data [67, 75, 48, 84, 74, 8, 78, 13]. Among them, learning and imposing heuristic style transmission strategies take the dominant positions, while explicitly separating and recombination images remain the boundary to explore. (4) *Meta-Learning.* Some of recent works [33, 39, 34, 17, 41, 14] simulate the distribution shifts between seen and unseen environments by using meta-learning [21], which splits the training data into meta-train and meta-test domains.

Despite a proliferation of DG approaches, Wiles *et al.* [71] reveal that the best DG methods are not consistent over different shifts and datasets in which disentangling offers limited improvements in most real-world scenarios. In addition, Gulrajani and Lopez-Paz [23] cast doubt on the progress that have been made by comparing to a standard empirical risk minimization baseline, showing that conventional domain-invariant methods [60, 22] exhibit robust improvements. In this work, we embrace the strengths of cross-domain invariance by considering a more fine-grained and stable property—*structural invariance*—which is evaluated on a number of challenging benchmarks.

**Data Mixing.**    Mixing data, such as Mixup [77, 66] and its various variants, has shown compelling results in standard supervised and semi-supervised learning. Mixup operations aim to conduct data interpolation via convex combinations of pairs of images and their labels. Instead, our CDM generates diverse training samples by replacing the background of a certain image with a randomly cropped patch from other images but keeps its object label fixed. CutMix [76] replaces a regularly-shaped image region with a patch from another training image and proportionally mix their ground-truth labels. CutPaste [31] cuts an image patch and randomly pastes it at an another image. BackErase [56] uses the object annotations to copy and pastes foreground objects on a background image. Compared to the aforementioned methods that cut a regularly-shaped image patch [76, 31] or rely on ground-truth object labels [56], the proposed CDM simultaneously leverage the category and domain information to derive an irregularly-shaped foreground regions while avoiding fine-grained annotations.

**Cross-Domain Invariance.**    Seeking cross-domain invariance is crucial for out-of-distribution generalization problems, such as Unsupervised Domain Adaptation (UDA) [49] and DG. Traditional methods typically resort to statistical distribution divergence, such as maximum mean discrepancy [43] and second-order moment [60], for measuring domain-wise distribution shifts. More recently, a promising approach is to utilize the concept of class prototype for enforcing semantic consistency across domains [73, 12, 10]. This line of research has also been extensively explored in various downstream tasks. However, we argue that no matter domain- or class-wise invariance cannot guarantee the generalizable representations especially when encountering unseen test environments.

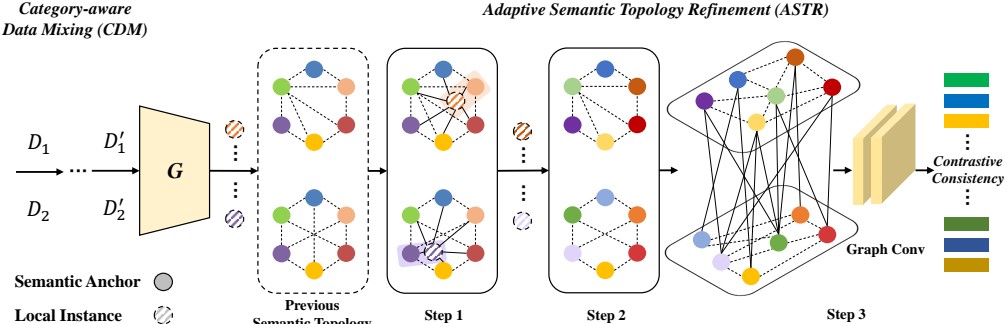

*Category-aware Data Mixing (CDM)*

*Adaptive Semantic Topology Refinement (ASTR)*

Figure 2: The pipeline of the proposed `MiRe`. Assume that we have two domains $\mathcal{D}_1$ and $\mathcal{D}_2$, CDM augments them to $\mathcal{D}_1'$ and $\mathcal{D}_2'$. In each iteration of ASTR, the embedding feature of each local instance performs feature aggregation based on the semantic topology in previous iteration and their local adjacent relations (**step 1**). Then, global anchor features are updated based on the aggregated local features (**step 2**). Finally, a bipartite graph learning procedure is imposed on the top of semantic topology to perform between-domain relational reasoning and induce structural invariance (**step 3**).

Such an invariance may be susceptible to include some misleading spurious correlation, such as wrongly associating an foreground object with specific background. The justification is that the topological structure of these prototypes is unexplored, making them being sensitive to the change of environments [44, 9, 79]. By contrast, the proposed `MiRe` models the graphical structures of different semantic anchors by means of feature aggregation and message-passing.

## 3 Methodology

In the task of DG, given that we have access to data from $N$ ($N \geq 2$) source domains $\mathcal{D}_s = \{\mathcal{D}_1, \mathcal{D}_2, ..., \mathcal{D}_N\}$ that are sampled from different data distributions defined on the joint space $\mathcal{X} \times \mathcal{Y}$. The $n$-th domain can be defined as $\mathcal{D}_n = \{(x_i, y_i, d_i)\}_{i=1}^{N_d}$, where $y_i = \{1, 2, ..., K\}$ is the class label, $d_i = \{1, 2, ..., N\}$ is the domain label, and $N_d$ is the number of samples in this domain. An overview of `MiRe` is illustrated in Fig. 2. `MiRe` consists of a CDM stage (Section 3.1) and an ASTR stage (Section 3.2), which jointly enable generalization on data-level and feature-level.

### 3.1 Category-aware Data Mixing

Most of DG works aim to learn predictive models with certain invariant properties that are independent of domain. As aforementioned, this sort of invariance may be brittle due to the presence of spurious correlation (cf. Fig. 1). To build robust classifier, many recent efforts are devoted to synthesize novel images/features through various augmentation strategies, such as image transformations [67] and style transformations [84, 48, 78]. They typically manipulate the low-level feature statistics (*e.g.,* color and texture) to generate diverse image styles while maintaining the content unchanged. However, we argue that these operations may still be constrained by the spurious correlation as style-agnostic representation is not sufficient to ensure semantic invariance [44].

Inspired by this, we introduce a novel data mixing strategy, which incorporates the categorical information to directly recombine source images. Considering that the category and domain labels of each sample are two orthogonal yet complementary supervision signals, we directly obtain the activation maps generated by these two classification networks for depicting the foreground regions.

An overview of the CDM is illustrated in Fig. 3. Specifically, for an input sample $x_i$, we first leverage the vanilla Grad-CAM [57] to generate activation maps $\mathcal{M}_c$ (class) and $\mathcal{M}_d$ (domain) by using its label $y_i$ and $d_i$, respectively. As shown in Fig. 3, the activation map ($\mathcal{M}_d$) induced by domain classification does not focus on the background as our expectation (this phenomenon holds in most DG benchmarks). Thus, we merge $\mathcal{M}_c$ and $\mathcal{M}_d$ to extract the foreground objects as follows,

$$\mathcal{M}_f := \texttt{threshold}(\mathcal{M}_c + \mathcal{M}_d), \tag{1}$$

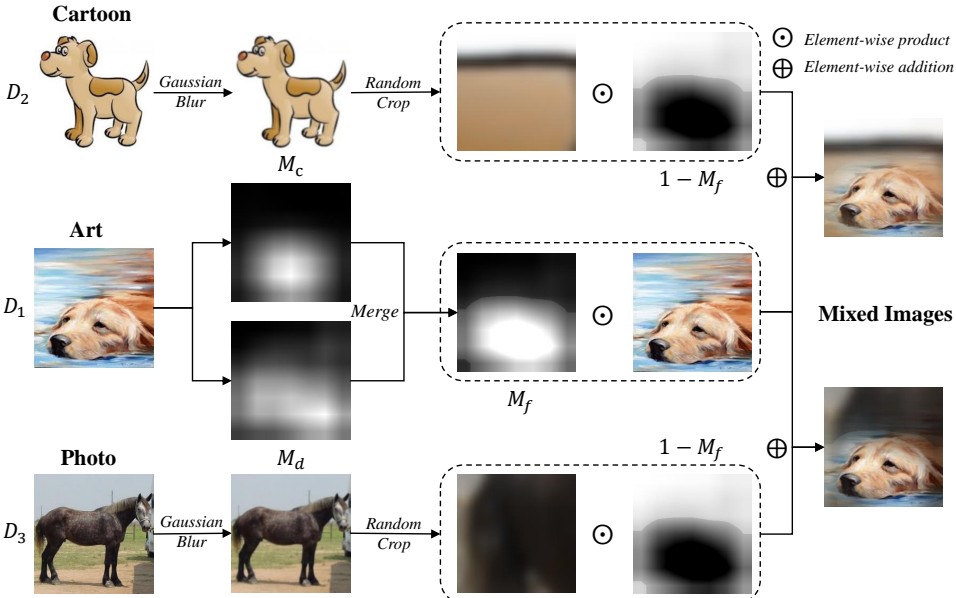

Figure 3: The workflow of CDM. For an image from Domain "Art", **(i)** extract two types of its foreground CAM ($\mathcal{M}_c$ and $\mathcal{M}_d$) and merge them ($\mathcal{M}_f$); **(ii)** randomly select two images from Domain "Cartoon" and Domain "Photo" to construct two backgrounds; **(iii)** apply Gaussian blur to the selected image and then randomly crop a patch from the blurred image; **(iv)** mix the foreground and background images by virtue of the computed $\mathcal{M}_f$.

where $\texttt{threshold}(\cdot)$ aims to reduce the outlier points and is set to 0.2 in all experiments to avoid additional tuning. Intuitively, $\mathcal{M}_d$ is a coarse-level supervision, which is sensitive to the domain-wise variations, while $\mathcal{M}_c$ is a fine-level supervision, which is sensitive to the intra-domain variations.

Next, to simulate the real-world situation that an object may appear in various scenes with distinct backgrounds, we randomly crop a small region of an image sampled from other domains. Before cropping, a Gaussian blur is applied to image $x_j$ to reduce image noise and detail, *i.e.,* $x'_j = \texttt{Gaussian}(x_j)$. A small patch is cropped from $x'_j$ and resized to the same scale, *i.e.,* $x_j^{\texttt{crop}} = \texttt{resize}(\texttt{crop}(x'_j))$. Finally, we synthesize a new image $x'_i$ by substituting the background of $x_i$ with the background of $x'_j$ using the merged Grad-CAM of image $x_i$ as a balancing weight, *i.e.,*

$$x_i^{\texttt{mix}} := \mathcal{M}_f \odot x_i + (1 - \mathcal{M}_f) \odot x_j^{\texttt{crop}}, \tag{2}$$

where $\odot$ is the element-wise product. The above operation is applied to every source image, and we will obtain $(N-1) \cdot |\mathcal{D}_s|$ images in all, where $|\mathcal{D}_s|$ is the number of original source images.

### 3.2 Adaptive Semantic Topology Refinement

**Structural Invariance.** After the data mixing, we introduce an Adaptive Semantic Topology Refinement (ASTR) mechanism to adaptively and progressively refine the topological structure of semantic anchors in the feature space, explicitly constraining the structural invariance across domains. The key idea is to endow the semantic anchor with the capability of reasoning over semantic topology globally, where semantic anchor is represented by the prototype features of each class. We argue that semantic topology, which consists of the semantic anchors as well as their interactions, is robust to unseen test environments. Technically, the structural invariance is characterised from two aspects: *cross-domain variation* and *cross-model variation* (historical and present models), implying that the semantic topology should be independent of the spatial and temporal changes.

**Overview.** We introduce an undirected graph $\mathcal{G} = (\mathcal{V}, \mathcal{E}, \boldsymbol{C})$ to represent the semantic topology, where $|\mathcal{V}| = K$ is the set of $K$ nodes and $\mathcal{E} \subseteq \mathcal{V} \times \mathcal{V}$ is the set of edges. The feature matrix is denoted by $\boldsymbol{C} = \{\boldsymbol{c}_1, \cdots, \boldsymbol{c}_K\} \in \mathbb{R}^{K \times D}$, where $\boldsymbol{c}_k$ denotes a prototype corresponding to the $k$-th class, $D$ is the feature dimension, and each element in $\boldsymbol{C}$ is associated to a node $\boldsymbol{v} \in \mathcal{V}$. The adjacency

matrix is denoted by $\boldsymbol{A} \in \mathbb{R}^{K \times K}$ where each element $\boldsymbol{A}_{ij}$ denotes the weight of edge $e_{ij} \in \mathcal{V}$. During training, each input sample is first compared to a set of semantic anchors and then adaptively aggregates features from its adjacent anchors to update its features. All updated sample features will be further used to update the corresponding anchor feature. Finally, the structural invariance is optimized based on bipartite graph learning.

**Step 0:** ASTR first computes the initial global anchor features $\boldsymbol{C}_{(0)}^{\texttt{global}}$ based on the vanilla DeepAll model, which is trained on both original and generated data. For the relation graph, we define its corresponding adjacency matrix $\mathcal{G}^{\texttt{global}}$ as follows,

$$\boldsymbol{A}_{ij}^{(I)} = \exp(-\frac{\|\boldsymbol{c}_i - \boldsymbol{c}_j\|_2^2}{2\sigma^2}), \tag{3}$$

where $I$ indicates the iteration times and $\sigma^2$ is the kernel bandwidth.

**Step 1:** At each training iteration, a batch of samples $\mathcal{B}$ is projected into the embedding space via the feature extractor $G$, *i.e.,* $\boldsymbol{f}_i = G(x_i), x_i \in \mathcal{B}$. For the sake of simplicity, we omit the subscript $i$. In the $I$-th iteration, for an image $x$ that belongs to class $k$, we perform feature aggregation as follows,

$$\boldsymbol{f}' = \phi \boldsymbol{f} + (1 - \phi) \sum_{j=1}^{K} w_j \cdot \boldsymbol{c}_j^{\texttt{global}}, \quad w_j = \frac{\exp((\boldsymbol{A}_{kj}^{(I-1)} + \cos(\boldsymbol{f}, \boldsymbol{c}_j))/2)}{\sum_j \exp((\boldsymbol{A}_{kj}^{(I-1)} + \cos(\boldsymbol{f}, \boldsymbol{c}_j))/2)}, \tag{4}$$

where $j = \{1, 2, ..., K\}$, $\cos(\cdot, \cdot)$ denotes cosine distance, and $\phi \in (0, 1)$ is a positive constant. $\cos(\boldsymbol{f}, \boldsymbol{c}_j)$ *locally* measures the relations between each instance and all semantic anchors, while $\boldsymbol{A}_{kj}^{(I-1)}$ provides the *global relations* among different semantic anchors. By doing so, each instance will be aware of the holistic semantic topology and dynamically conduct aggregation.

**Step 2:** Then, we update the global anchor features based on $\mathcal{B}$ in a moving average manner,

$$c_{j(I)}^{\texttt{global}} = \phi \, c_{j(I)}^{\texttt{local}} + (1 - \phi) \, c_{j(I-1)}^{\texttt{global}}, \tag{5}$$

where $c_{j(I)}^{\texttt{local}}$ stands for the local anchor features computed from all $\boldsymbol{f}'$. Meanwhile, the adjacency matrix will be updated accordingly.

**Step 3:** Finally, we optimize the cross-domain and cross-model consistencies to achieve structural invariance. A naive approach is to compare each anchor in one domain/model to all anchors in the other domain/model. However, the structural relations of each anchor within the graph are neglected. On the other hand, the constructed graph may still contain some spurious edges, leading to unreliable inter-graph node correspondence reasoning. Remedying these issues, we introduce a Bipartite Graph Convolutional Network (BGCN) to perform cross-domain relational reasoning based on a bipartite graph $\mathcal{G}^{\texttt{bg}}(\mathcal{V}_{s'}, \mathcal{V}_{s''}, \mathcal{E}, \boldsymbol{A}_{ij}^{\texttt{bg}})$, where $s'$ and $s''$ can be any two source domains. The node affinities between $\mathcal{V}_{s'}$ and $\mathcal{V}_{s''}$ are simultaneously constrained by cross-domain and cross-model variations. Specifically, for a node $v_i$ in $\mathcal{V}_{s'}$ and a node $v_j$ in $\mathcal{V}_{s''}$, we have,

$$\boldsymbol{A}_{ij}^{\texttt{bg}} = \frac{\cos(\boldsymbol{c}_{i(I-1)}^{\texttt{global}}, \boldsymbol{c}_{i(I)}^{\texttt{global}}) + \cos(\boldsymbol{c}_{j(I-1)}^{\texttt{global}}, \boldsymbol{c}_{j(I)}^{\texttt{global}})}{2} \cdot \cos(\boldsymbol{c}_{i(I)}^{\texttt{global}}, \boldsymbol{c}_{j(I)}^{\texttt{global}}), \tag{6}$$

By doing so, the influence of those unreliable intra- and inter-graph correspondence were largely reduced. Then, we conduct bipartite graph convolution [28, 11] over $\mathcal{G}^{\texttt{bg}}$ by stacking multiple layers. Meanwhile, we introduce a contrastive consistency regularization on the top of BGCN,

$$\mathcal{L}_{\text{CCR}} = \sum_{k=1}^{K} \Phi(z_{\mathcal{V}_{s'}}^k, z_{\mathcal{V}_{s''}}^k) + \sum_{m \neq n} (\max\{0, \xi - \Phi(z_{\mathcal{V}_{s'}}^m, z_{\mathcal{V}_{s''}}^n)\}), \tag{7}$$

where $z$ is the embedded vector through BGCN, $\Phi$ is the squared Euclidean distance, and $\xi$ is the margin term and is set to 2 in experiments. To this end, the overall training objective of MiRe is,

$$\min_{\theta} \mathbb{E}_{(x,y) \sim \mathcal{P}} \, \mathcal{L}_{\text{cls}} + \lambda \cdot \mathcal{L}_{\text{CCR}}, \tag{8}$$

where $\mathcal{L}_{\text{cls}}$ is the standard cross-entropy loss and $\lambda$ is the trade-off parameter.

Table 1: Generalization results (%) on PACS benchmark.

| Source | Target | DeepAll - | JiGen [6] | CrossGrad [59] | MSAF [17] | ER [80] | Metareg [4] | RSC [25] | MixStyle [84] | SagNet [48] | EFDM [78] | MiRe (Ours) |
|---|---|---|---|---|---|---|---|---|---|---|---|---|
| C,P,S | Art | 77.0±0.3 | 79.4 | 79.8 | 80.3 | 80.7 | 83.7 | 78.9 | 83.1 | 83.6 | 83.9 | 84.6±0.5 |
| A,P,S | Cartoon | 74.8±0.5 | 75.3 | 76.8 | 77.2 | 76.4 | 77.2 | 76.9 | 78.6 | 77.7 | 79.4 | 79.5±0.4 |
| A,C,S | Photo | 95.8±0.1 | 96.0 | 96.0 | 95.0 | 96.7 | 95.5 | 94.1 | 95.9 | 95.5 | 96.8 | 96.8±0.2 |
| A,C,P | Sketch | 70.0±0.6 | 71.6 | 70.2 | 71.7 | 71.8 | 70.3 | 76.8 | 74.2 | 76.3 | 75.0 | 78.4±1.0 |
| Average | | 79.4 | 80.5 | 80.7 | 81.1 | 81.4 | 81.7 | 81.7 | 82.9 | 83.3 | 83.9 | **84.8** |

Table 2: Generalization results (%) on VLCS benchmark.

| Source | Target | DeepAll - | DBA-DG [32] | MMD-AAE [36] | MLDG [33] | Epi-FCR [34] | JiGen [6] | MASF [17] | MMLD [45] | SFA-A [35] | MiRe (Ours) |
|---|---|---|---|---|---|---|---|---|---|---|---|
| L,C,S | VOC | 71.8±0.6 | 70.0 | 67.7 | 67.7 | 67.1 | 70.6 | 69.1 | 72.0 | 70.4 | 70.3±0.3 |
| V,C,S | LabelMe | 61.1±0.4 | 63.5 | 62.6 | 61.3 | 64.3 | 60.9 | 64.9 | 58.8 | 62.0 | 63.6±0.7 |
| V,L,S | Caltech | 95.8±0.2 | 93.6 | 94.4 | 94.4 | 94.1 | 96.9 | 94.8 | 96.7 | 97.2 | 96.2±0.4 |
| V,L,C | Sun | 62.5±0.6 | 61.3 | 64.4 | 64.4 | 65.9 | 64.3 | 67.6 | 68.1 | 66.2 | 69.4±0.7 |
| Average | | 72.8 | 72.1 | 72.3 | 72.3 | 72.9 | 73.2 | 74.1 | 73.9 | 74.0 | **74.9** |

Table 3: Generalization results (%) on Office-Home benchmark.

| Source | Target | DeepAll - | MMD-AAE [36] | CrossGrad [59] | JiGen [6] | DDAIG [82] | DOSN [58] | RSC [25] | SagNet [48] | MixStyle [84] | MiRe (Ours) |
|---|---|---|---|---|---|---|---|---|---|---|---|
| C,P,R | Art | 59.4±0.4 | 59.9 | 58.4 | 53.0 | 59.2 | 59.4 | 58.4 | 60.2 | 58.7 | 60.2±0.8 |
| A,P,R | Clipart | 48.0±1.1 | 47.3 | 49.4 | 47.5 | 52.3 | 45.7 | 47.9 | 45.4 | 53.4 | 53.2±0.9 |
| A,C,R | Product | 72.7±0.5 | 72.1 | 73.9 | 71.5 | 74.6 | 71.8 | 71.6 | 70.4 | 74.2 | 75.1±0.6 |
| A,C,P | Real | 75.3±0.4 | 74.8 | 75.8 | 72.8 | 76.0 | 74.7 | 74.5 | 73.4 | 75.9 | 76.4±0.6 |
| Average | | 63.9 | 62.7 | 64.4 | 61.2 | 65.5 | 62.9 | 63.1 | 62.4 | 65.5 | **66.2** |

## 4 Experiments

### 4.1 Setup

**Datasets.** We fully verify the generalization performance of MiRe on four standard DG benchmarks: **PACS** [32], **VLCS** [19], **Office-Home** [65], and **DomainNet** [52]. (1) PACS is the most-widely used DG benchmark, which has 9,991 images of 7 classes from four kinds of environment: *Photo*, *Art Painting*, *Cartoon*, and *Sketch*. (2) VLCS contains 10,729 images of 5 classes from four photographic domains: *PASCAL VOC 2007* [18], *LabelMe* [55], *Caltech* [20], and *Sun* [72]. (3) Office-Home is composed of 15,500 images of 65 classes from four domains (*Artistic*, *Clipart*, *Product*, and *Real World*). The images are collected from office and home environments. (4) DomainNet is a more recent large-scale dataset for multi-source domain adaptation and DG. It spans 0.6 million images of 345 classes from six domains (*Clipart*, *Infograph*, *Quickdraw*, *Painting*, *Real* and *Sketch*).

**Model Selection.** To facilitate fair comparisons, we strictly follow the model selection strategy used by baseline methods. In general, we follow the leave-one-domain-out setting, *i.e.,* one domain is chosen as the held out domain and the remaining domains are seen as source domains. For PACS, we follow the original train and val splits established by [32]. On VLCS, according to the prior methods [6, 45], we split 30% of the source samples as validation datasets. For Office-Home, we employ the train and val splits established in [32, 83, 84]. On DomainNet, we follow the train and val splits used in [7], *i.e.,* randomly divide the train split from [52] in a 90-10% proportion to obtain train and val splits. Then, we select the model that shows the best performance in the validation dataset.

**Implementation Details.** For PACS, Office-Home, and DomainNet, we use ResNet-18 [24] pre-trained on the ImageNet as the backbone feature extractor. To include more baselines for comparison, we use AlexNet [30] for VLCS. On DomainNet, we also provide results of ResNet-50. The networks are trained using SGD with momentum of 0.9 and weight decay of 5e-4 for 100 epochs. For PACS, VLCS, and Office-Home, the batch size is set to 16. For DomainNet, the batch size is set to 256. For CDM, the cropped area ratio (background image) is empirically set to $\frac{1}{8}$. For Eq. (4), we set $\phi = 0.5$. For Eq. (8), we set $\lambda = 0.1$. We evaluate the out-of-domain generalization performance in all object

Table 4: Generalization results (%) on DomainNet benchmark.

| | Method | Clipart | Infograph | Quickdraw | Painting | Real | Sketch | Avg |
|---|---|---|---|---|---|---|---|---|
| **ResNet-18** | DeepAll | 57.2 | 17.7 | 43.2 | 13.9 | 54.9 | 39.4 | 37.7 |
| | Multi-Headed [7] | 55.4 | 17.5 | 40.8 | 11.2 | 52.9 | 38.6 | 36.1 |
| | MetaReg [4] | 53.6 | 21.0 | 45.2 | 10.6 | 58.4 | 42.3 | 38.5 |
| | DMG [7] | 60.0 | 18.7 | 44.5 | 14.1 | 54.7 | 41.7 | 39.0 |
| | MiRe | 62.5 | 23.0 | 46.5 | 15.7 | 59.1 | 44.4 | **41.9** |
| **ResNet-50** | DeepAll | 61.8±0.6 | 20.2±0.7 | 45.0±1.3 | 14.3±1.0 | 57.3±0.7 | 44.8±0.6 | 40.6 |
| | Multi-Headed [7] | 61.7 | 21.2 | 46.8 | 13.8 | 58.4 | 45.4 | 41.2 |
| | MetaReg [4] | 59.7 | 25.5 | 50.1 | 11.5 | 64.5 | 50.0 | 43.6 |
| | DMG [7] | 65.2 | 22.1 | 50.0 | 15.6 | 59.6 | 49.0 | 43.6 |
| | MiRe | 64.7±1.0 | 27.8±1.2 | 53.1±0.8 | 16.4±1.5 | 64.1±0.6 | 52.3±0.9 | **46.4** |

classes and report the means and standard derivations over 10 runs with different random seeds. We implement all experiments based on the PyTorch framework.

## 4.2 Baselines

We compare MiRe against state-of-the-art DG methods, which are summarized as follows. **DeepAll:** Training a classification model through simply merging all source domains. **Feature Alignment:** MMD-AAE [36] and EFDM [78]. **Feature Disentanglement:** DOSN [58], DMG [7] and Sag-Net [48]. **Data Augmentation:** JiGen [6] and DDAIG [82]. **Feature Augmentation:** Cross-Grad [59], MixStyle [84] and SFA-A [35]. **Meta-Learning:** MASF [17] and Metareg [4]. **Other SOTA:** RSC [25] designs a model-agnostic training strategy, and ER [80] ensures the conditional invariance via an entropy regularization term. For most of the compared approaches, we cite the classification accuracy reported in their original papers. In PACS, the results of 'RSC' and 'MixStyle' are obtained with their official codes by following the leave-one-domain-out setting.

## 4.3 Results

**PACS and VLCS.** In Tab. 1 and Tab. 2, we display the quantitative comparisons with baseline methods on PACS and VLCS benchmarks. As can be seen, MiRe outperforms state-of-the-art DG methods by large margins, revealing the effectiveness of the proposed structural invariance in real-world data. In particular, MiRe improves over the best baseline method [78] (CVPR 2022) by 0.9% on the PACS dataset. For the hard generalization task A, C, S → Sketch, our method substantially outperforms the suboptimal result by 1.6%.

**Office-Home and DomainNet.** Tab. 3 and Tab. 4 report the results on Office-Home and DomainNet benchmarks, which have larger data volume in contrast to PACS and VLCS. The proposed MiRe substantially and consistently exceeds all comparison methods, showing the scalability of our MiRe on large-scale datasets with distinct classes.

## 4.4 Empirical Analyses and Ablations

**Ablation Study.** We investigate the role of each part of MiRe by conducting in-depth ablation studies on four benchmarks. The results are listed in Tab. 5. We highlight two sets of key observations. (1) CDM w/o $(1 - \mathcal{M}_d)$ represents that we replace $\mathcal{M}_d$ by $(1 - \mathcal{M}_d)$ in Eq. (1). The results testify that the domain classification evidence is beneficial for depicting the foregrounds. In addition, $\mathcal{M}_c$, $\mathcal{M}_d$, and Gaussian blur operation are reasonably designed, providing high-quality augmentation results. Only with CDM could provides very competitive results. (2) Introducing relation graphs to model semantic topology plays an important role in encouraging the so-called structural invariance especially on the challenging benchmark, such as DomainNet.

**The Effects of CDM.** Since CDM does not requires any specific model architectures and training procedures, we apply it to three existing methods, *i.e.,* ER [80], MixStyle [84] and EFDM [78], to further enhance their classification performance. The results are shown in Tab. 6, where CDM consistently improves over the baseline methods.

Table 5: Ablation of `MiRe` on four DG benchmarks (%).

| Method | PACS | VLCS | Office-Home | DomainNet | Average |
|---|---|---|---|---|---|
| DeepAll | 79.4 | 72.8 | 63.9 | 37.7 | 63.5 |
| `MiRe` w/o CDM | 83.5 | 74.0 | 65.1 | 39.3 | 65.5 |
| CDM w/o $\mathcal{M}_c$ | 82.6 | 72.8 | 65.0 | 39.4 | 65.0 |
| CDM w/o $\mathcal{M}_d$ | 84.4 | 74.4 | 65.7 | 41.0 | 66.4 |
| CDM w/o Gaussian Blur | 83.9 | 74.2 | 64.6 | 40.6 | 65.8 |
| CDM w/ $(1 - \mathcal{M}_d)$ | 83.0 | 73.4 | 64.9 | 38.8 | 65.0 |
| `MiRe` w/o ASTR | 82.3 | 71.8 | 64.7 | 38.9 | 64.4 |
| ASTR w/o Cross-Domain Invariance | 84.0 | 74.5 | 65.6 | 40.6 | 66.2 |
| ASTR w/o Cross-Model Invariance | 84.3 | 74.6 | 65.7 | 41.0 | 66.4 |
| ASTR w/o Graph Structure | 82.8 | 73.3 | 64.8 | 40.1 | 65.3 |
| ASTR w/o Feature Aggregation | 84.2 | 74.1 | 65.7 | 40.4 | 66.1 |
| `MiRe` | 84.8 | 74.9 | 66.2 | 41.9 | 67.0 |

Table 6: The effects of CDM on PACS benchmark.

| Source | Target | DeepAll | CDM | + ER (81.4) | + MixStyle (82.9) | + EFDM (83.9) |
|---|---|---|---|---|---|---|
| C,P,S | Art | 77.0 | 80.8 | 80.4 | 82.9 | 83.8 |
| A,P,S | Cartoon | 74.8 | 78.5 | 77.8 | 78.2 | 81.3 |
| A,C,S | Photo | 95.8 | 94.8 | 94.4 | 96.0 | 95.5 |
| A,C,P | Sketch | 70.0 | 75.0 | 77.4 | 77.6 | 78.2 |
| Average | | 79.4 | 82.3 | 82.5 (+1.1) | 83.7 (+0.8) | 84.7 (+0.8) |

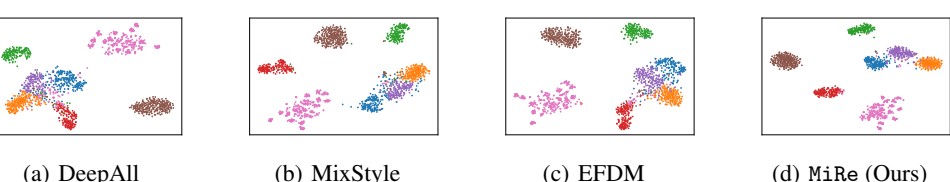

(a) DeepAll  (b) MixStyle  (c) EFDM  (d) `MiRe` (Ours)

Figure 4: The t-SNE visualization of extracted deep features using different methods on PACS dataset. The different colors stands for different classes.

Table 7: Results (%) on Chest X-Rays data.

| Method | RSNA | ChexPert | NIH |
|---|---|---|---|
| ERM | 55.1 | 60.9 | 53.4 |
| IRM | 57.0 | 63.3 | 54.6 |
| CSD | 58.6 | 64.4 | 54.7 |
| MatchDG | 58.2 | 59.0 | 53.2 |
| ASTR | **63.6** | **65.0** | **56.4** |

**The Effects of ASTR.** To testify the robustness of ASTR under the presence of spurious correlation, we additionally perform experiments on three Chest X-ray datasets: NIH [70], ChexPert [27] and RSNA [1]. We follow the same settings as in [44] to create spurious correlation. The empirical comparisons are shown in Tab. 7, where the results of baseline methods (*i.e.,* ERM [64], IRM [3], CSD [53] and MatchDG [44]) are cited from [44].

**Feature Visualization.** We use the t-SNE algorithm [63] to visualize the deep representations in ResNet-18 when *Photo* (PACS benchmark) is regarded as target domain. We compare the proposed `MiRe` with DeepAll, MixStyle [84], and EFDM [78] in Fig. 4. As expected, `MiRe` exhibits better clustering (class) patterns due to the refinement of semantic topology during training.

## 5    Conclusion

In this paper, we proposed the `MiRe` framework to achieve out-of-distribution generalization. The key idea of our method is to endow the predictive models with the capability of reasoning over the semantic topology. `MiRe` instantiates this objective with the incorporation of two elaborate modules, *i.e.,* CDM and ASTR. CDM augments the original data to reduce potential bias towards spurious correlations, while ASTR builds relation graphs to represent semantic topology, which is updated through feature aggregation and message-passing. Experiments on standard benchmarks verified the effectiveness of `MiRe`. Additional empirical analysis reveals the individual effect of each component.

## Acknowledgements

This work was supported by Hong Kong Research Grants Council through Research Impact Fund (Grant R-5001-18).

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
