# Mix and Reason: Reasoning over Semantic Topology with Data Mixing for Domain Generalization (Supplementary Material)

**Chaoqi Chen**[1]  **Luyao Tang**[2]  **Feng Liu**[3]  **Gangming Zhao**[1]  **Yue Huang**[2]  **Yizhou Yu**[1]*
[1]The University of Hong Kong  [2]Xiamen University  [3]Deepwise AI Lab
cqchen1994@gmail.com, lytang@stu.xmu.edu.cn, liufeng@deepwise.com
gmzhao@connect.hku.hk, yhuang2010@xmu.edu.cn, yizhouy@acm.org

## A  Training Procedure of Mix and Reason (`MiRe`)

Algorithm 1 depicts the complete training procedure of Mix and Reason (`MiRe`). To facilitate a better understanding, we further highlight three aspects regarding our training process.

- The semantic topology is a domain-agnostic concept. In practice, we first instantiate this concept in each domain via the relation graph. Then, we enforce its robustness by encourage the invariance of topological relations among different graphs. As we always uses mini-batch SGD for optimization, the mini-batch samples will progressively refine the intra-domain relation graph and inter-domain structural invariance.

- The word 'reasoning' in our paper indicates that we perform intra-domain and inter-domain relational reasoning via the constructed graphs and their corresponding learning process.

- In Eq. (3), we build a *intra-domain* relation graph based on the whole data from one domain. This relation graph is iteratively updated by samples in each mini-batch based on feature aggregation defined in Eq. (4). In Eq. (6), we build a *inter-domain* bipartite graph based on the whole data from any two different domains. This bipartite graph is updated by a contrastive consistency loss (*i.e.* Eq. (7)). In that sense, the intra-domain and inter-domain relation graphs are cascaded, where the intra-domain graph perceives *the topological structure in each domain* while inter-domain relation graph ensures the *structural invariance* among different domains.

## B  Additional Discussion

**Motivation of CDM.** In real-world datasets, we assume that each category may have distinct contexts [18], namely, hybrid context generalization. In that sense, the activation maps $\mathcal{M}_d$ generated by domain classification loss might not utilize the random and unique contexts (differ in each category or even each instance) as their classification evidences. By contrast, we observe that $\mathcal{M}_d$ focuses on the foreground regions, which characterize the dominant difference between domains. For example, in PACS datasets, the major difference among these domains is that the foreground objects are collected from various styles (object style instead of whole image style), which is crucial for semantically characterizing a category. The activation maps $\mathcal{M}_c$ generated by object classification helps to achieve foreground and background separation, while $\mathcal{M}_d$ helps to enhance the completeness of depicting foregrounds. Hence, we choose to merge these two activation maps in practice.

---

*Corresponding author

36th Conference on Neural Information Processing Systems (NeurIPS 2022).

**Algorithm 1** Training Procedure of Mix and Reason (`MiRe`), $|\mathcal{D}_n|$ denotes number of samples in the $n$-th domain, $I$ is the iteration times, $\mathcal{B}$ is the mini-batch training set.

---

**Require:** $N$ ($N \geq 2$) source domains $\mathcal{D}_s = \{\mathcal{D}_1, \mathcal{D}_2, ..., \mathcal{D}_N\}$
 1: *Category-aware Data Mixing (CDM):*
 2: Training a class predictor and a domain predictor based on $\mathcal{D}_s$
 3: **for** $n = 1$ **to** $N$ **do**
 4:  **for** $i = 1$ **to** $|\mathcal{D}_n|$ **do**
 5:    Compute $\mathcal{M}_c$ based on the class predictor and $\mathcal{M}_d$ based on the domain predictor
 6:    Merging $\mathcal{M}_c$ and $\mathcal{M}_d$ by Eq. (1), output: $\mathcal{M}_f$
 7:    **for** $j = 1$ **to** $N$ ($j \neq n$) **do**
 8:      Randomly select an image $x_j$ from domain $\mathcal{D}_j$
 9:      Apply Gaussian blur to $x_j$, output: $x'_j$
10:      Randomly crop a patch from $x'_j$ and resize it to the same scale of $x_j$, output: $x_j^{\mathtt{crop}}$
11:      Mix $x_i$ and $x_j^{\mathtt{crop}}$ by Eq. (2), output: $x_i^{\mathtt{mix}}$
12:    **end for**
13:  **end for**
14: **end for**
15: Output the augmented source domains $\hat{\mathcal{D}}_s$
16: *Adaptive Semantic Topology Refinement (ASTR):*
17: Train a vanilla DeepAll model based on $\hat{\mathcal{D}}_s$
18: Compute the initial global anchor features $C_{(0)}^{\mathtt{global}}$ *in each domain* based on the DeepAll model
19: Compute the initial adjacency matrix in each domain by Eq. (3)
20: **for** $I = 1$ **to** `MaxIter` **do**
21:  Randomly and uniformly sample a batch of samples $\mathcal{B}$ from $\hat{\mathcal{D}}_s$
22:  **for** $i = 1$ **to** $|\mathcal{B}|$ **do**
23:    Perform feature aggregation by Eq. (4)
24:    Update the corresponding global anchor feature by Eq. (5)
25:  **end for**
26:  Update the adjacency matrix by Eq. (3) based on the updated global anchor features
27:  Build *inter-domain* bipartite graphs by Eq. (6)
28:  Compute the contrastive consistency loss by Eq. (7)
29:  Compute the overall loss by Eq. (8)
30: **end for**

---

# C  Additional Experiments

## C.1  DomainBed

DomainBed [6] is a newly established testbed for domain generalization including seven multi-domain datasets and fourteen baseline algorithms. Considering that the evaluation protocol established by DomainBed is computationally expensive, which requires about 4,142 models for per DG algorithm. Hence, we choose the five real-world and challenging datasets as the main evaluation benchmarks, *i.e.,* PACS, VLCS, Office-Home, TerraInc and DomainNet.

The baseline algorithms can be divided into five categories: **(i) Empirical Risk Minimization:** Standard Empirical Risk Minimization (ERM) [20]; **(ii) Domain-Specific Representation Learning:** Group Distributionally Robust Optimization (GroupDRO) [17], Marginal Transfer Learning (MTL) [2], Adaptive Risk Minimization (ARM) [22]; **(iii) Meta-learning:** Meta-Learning for DG (MLDG) [9]; **(iv) Cross-Domain Invariance:** Invariant Risk Minimization (IRM) [1], Deep Correlation Alignment (CORAL) [19], Maximum Mean Discrepancy (MMD) [11], Domain Adversarial Neural Networks (DANN) [5], Class-conditional DANN (CDANN) [13], Variance Risk Extrapolation (VREx) [8]; **(v) Augmentation:** Interdomain Mixup (Mixup) [21], Representation Self Challenging (RSC) [7], Style-Agnostic Networks (SagNet) [14].

Noting that we reproduce the results of ERM [20] (*i.e.,* DeepAll in main paper) in our experiments for fair comparisons. Following evaluation protocols established in [6], we report results using training domain as validation set for model selection. The experimental results are shown in Table 1, where

we can observe that our `MiRe` consistently outperforms all baseline methods, demonstrating the versatility and robustness of `MiRe` on different DG benchmarks and evaluation protocols.

Table 1: Domain generalization results (%) on DomainBed [6] benchmark. We highlight the **best results** and the second best results. The backbone network is ResNet-50.

| Method | PACS | VLCS | Office-Home | TerraInc | DomainNet | Average |
|---|---|---|---|---|---|---|
| ERM [20] | 85.5 | 77.5 | 66.5 | 46.1 | 40.9 | 63.3 |
| ERM (reproduced) | 84.5 | 77.2 | 66.6 | 45.9 | 40.1 | 62.9 |
| IRM [1] | 83.5 | 78.6 | 64.3 | 47.6 | 33.9 | 61.6 |
| GroupDRO [17] | 84.4 | 76.7 | 66.0 | 43.2 | 33.3 | 60.7 |
| Mixup [21] | 84.6 | 77.4 | 68.1 | 47.9 | 39.2 | 63.4 |
| MLDG [9] | 84.9 | 77.2 | 66.8 | 47.8 | 41.2 | 63.6 |
| CORAL [19] | 86.2 | **78.8** | 68.7 | 47.7 | 41.5 | 64.5 |
| MMD [11] | 84.7 | 77.5 | 66.4 | 42.2 | 23.4 | 58.8 |
| DANN [5] | 83.7 | 78.6 | 65.9 | 46.7 | 38.3 | 62.6 |
| CDANN [13] | 82.6 | 77.5 | 65.7 | 45.8 | 38.3 | 62.0 |
| ARM [22] | 85.1 | 77.6 | 64.8 | 45.5 | 35.5 | 61.7 |
| RSC [7] | 85.2 | 77.1 | 65.5 | 46.6 | 38.9 | 62.7 |
| MTL [2] | 84.6 | 77.2 | 66.4 | 45.6 | 40.6 | 62.9 |
| SagNet [14] | 86.3 | 77.8 | 68.1 | 48.6 | 40.3 | 64.2 |
| VREx [8] | 84.9 | 78.3 | 66.4 | 46.4 | 33.6 | 61.9 |
| `MiRe` (Ours) | **87.2** | 78.4 | **69.5** | **49.3** | **45.8** | **66.0** |

## C.2 Semantic Segmentation

To additionally investigate the out-of-distribution generalization performance of the formulated *structural invariance*, we conduct experiments on a challenging out-of-domain segmentation task (GTA5 $\rightarrow$ Cityscapes), *i.e.,* train a segmentation model on GTA5 and directly test it on Cityscapes. GTA5 [16] is a synthetic dataset generated from Grand Theft Auto 5 game engine. Cityscapes [4] is a real-world dataset collected from different cities in primarily Germany. Following the common practice of previous cross-domain semantic segmentation methods [10], we utilize DeepLab-v2 [3] segmentation network with ResNet101 backbone. Mean Intersection over Union (mIOU) and mean Accuracy (mAcc) of all object categories are used for evaluation. We remove the CDM module and bipartite graph learning head from `MiRe` and plug the rest components into the segmentation model, *i.e.,* `MiRe`$^*$.

Table 2: Experiment results of semantic segmentation from GAT5 to Cityscapes.

| Method | Reference | mIOU (%) | mAcc (%) |
|---|---|---|---|
| DeepAll | - | 37.0 | 51.5 |
| pAdaIN [15] | CVPR 2021 | 38.3 | 52.1 |
| Mixstyle [23] | ICLR 2021 | 40.3 | 53.8 |
| DSU [12] | ICLR 2022 | 43.1 | 57.0 |
| `MiRe`$^*$ | Ours | **44.0** | **58.5** |

Table 2 displays the quantitative segmentation results on Cityscapes. Figure 1 provides qualitative segmentation results on Cityscapes. As can be seen, the property of structural invariance is robust to different scenarios and significantly outperforms the baseline methods.

## C.3 Visualization

We visualize the relations (edge weights) of semantic anchors (nodes) on the PACS dataset. As shown in figure 2, we can observe that those semantically similar anchors will be assigned with larger weights, while those dissimilar anchors will be assigned with small weights. More importantly, such relations will be generalizable across domains, fitting the human intelligence that humans are talented at comparing and reasoning when learning new concepts.

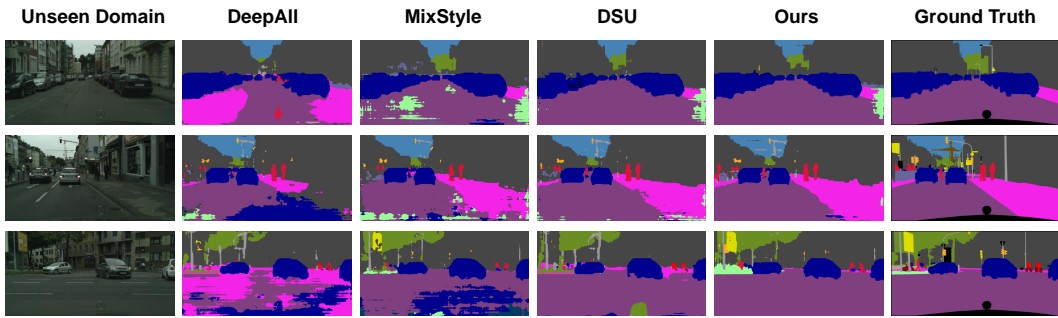

Figure 1: The visualization results of different DG methods on Cityscapes (unseen domain).

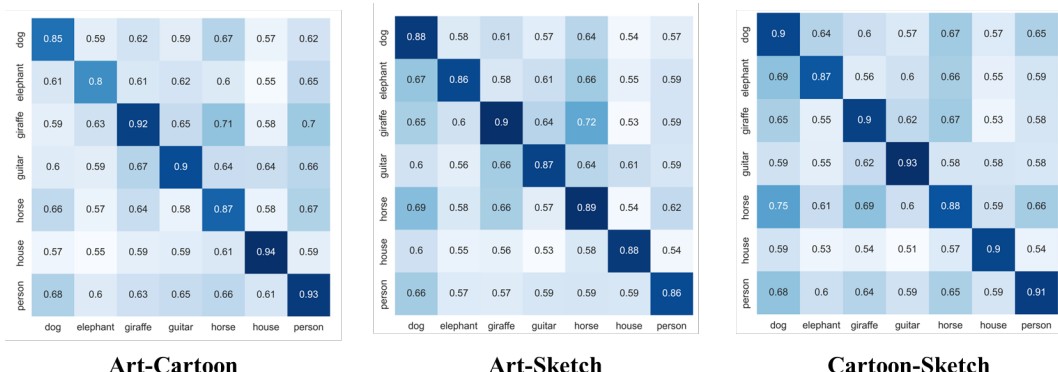

Figure 2: The visualization results of edge weights on the PACS dataset.

## C.4 Hyperparameters

We respectively testify the robustness of three hyperparameters (*i.e.,* the threshold in Eq. (1), $\xi$ in Eq. (7) and $\lambda$ in Eq. (8)) on the PACS and Office-Home datasets. The results are shown in Table 3, Table 4, and Table 5. From the table, we can see that the out-of-domain generalization performance of our method is relatively insensitive to the variations of these hyperparameters.

Table 3: The robustness of threshold in Eq. (1).

| Threshold | PACS | Office-Home | Avg |
|:---:|:---:|:---:|:---:|
| 0.05 | 83.8 | 65.9 | 74.9 |
| 0.10 | 84.2 | 66.1 | 75.2 |
| 0.15 | 84.5 | 66.0 | 75.3 |
| 0.20 | 84.8 | 66.2 | 75.5 |
| 0.25 | 84.7 | 66.2 | 75.5 |
| 0.30 | 84.4 | 66.1 | 75.3 |
| 0.35 | 84.0 | 65.5 | 74.8 |

Table 4: The robustness of $\xi$ in Eq. (7).

| $\xi$ | PACS | Office-Home | Avg |
|-----|------|-------------|------|
| 0.3 | 83.5 | 65.8 | 74.7 |
| 0.7 | 84.6 | 66.0 | 75.3 |
| 1.0 | 84.8 | 66.2 | 75.5 |
| 1.5 | 84.5 | 66.1 | 75.3 |
| 2.0 | 84.1 | 65.7 | 74.9 |

Table 5: The robustness of $\lambda$ in Eq. (8).

| $\lambda$ | PACS | Office-Home | Avg |
|------|------|-------------|------|
| 0.01 | 83.4 | 65.7 | 74.6 |
| 0.05 | 84.0 | 66.0 | 75.0 |
| 0.08 | 84.7 | 66.2 | 75.5 |
| 0.10 | 84.8 | 66.2 | 75.5 |
| 0.12 | 84.8 | 66.1 | 75.5 |
| 0.15 | 84.6 | 66.0 | 75.3 |
| 0.20 | 84.3 | 65.6 | 75.0 |