# OpenReview forum: "Mix and Reason: Reasoning over Semantic Topology with Data Mixing for Domain Generalization"
_NeurIPS.cc/2022/Conference — NeurIPS 2022 Accept_

### Official Review · Reviewer_Ywq6 · 2022-06-23

**Rating:** 4
**Confidence:** 5
**Soundness:** 2 fair
**Presentation:** 2 fair
**Contribution:** 2 fair

**Summary:**

This paper introduces a method called MiRe, for Mix and reason over semantic topology, to do domain generalization. The motivation is that existing work typically ignores the semantic information of the inputs and further lack topological structure, which authors claim to be consistent across domains. MiRe consists of two steps: background (other domains) + foreground (current domain) Mixup to generate new samples in different backgrounds, and graph network to mine the neighbor class feature information. Experiments are done in various benchmark datasets and demonstrates the efficacy of MiRe.

=======

Post rebuttal: I really appreciate authors' efforts in providing detailed rebuttal, which resolves most of my concerns. I hope that the future version can include the modifications w.r.t. all comments. But please understand that everyone has a standard for accepting a paper. To me, my main concern is that the novelty is not that "sparking" and the method is extremely complicated with so many hyperparameters to tune. Thus, I could not give you high scores. However, I will not stand against rejection. I will give you a 4 accordingly. I'm trying to be nice to you even if my papers still do not get any responses...

**Questions:**

See above comments.

**Strengths And Weaknesses:**

### Strength

1. The idea of bringing Mixup operation to domain generalization and further enhance its semantic representation is interesting.
2. The writing is sound, with interesting figures to show the ideas.
3. The experiments are extensive, with different comparison methods.

### Weakness

#### 1. Overclaims or several claims are not evaluated.

There are many claims in this paper and I fail to find supports for some of them. Thus, this paper could have some overclaims:
- In L36, how to evaluate that MiRe does not have the "data-dependent'' spurious correlations? I fear that MiRe is still a data-driven method and it heavily rely on the Mixup operation of the fore-background.
- In L42, authors claimed that previous methods require "some distribution of values for an attribute". I would assume MiRe does not. However, the adoption of Grad-Cam indeed assumes that the input data is image with clear fore and background information. Thus, it is an overclaim.
- In L44, I fail to find the answers to the question: "how many factors of the latent factors".
- In L132, why do existing data augmentation approaches lack diversity? Any support?

#### 2. Methodology.

- Technically, the method is not novel. It is a combination of Grad-Cam, Mixup, and GCN. I admire the application of these existing techniques. But there is no further insight and motivation is not strong. Plus, there are the following issues:
- In figure 3, why only cropping 1/8 of the original image? There lacks motivation.
- Similarly, in Eq. 4, why adopting $\operatorname{cos}(f, c_j) / 2$? How did the term $1/2$ come up? This also goes to Eq. 6.
- The second part of the method is too complicated and computationally expensive for multiple domains. Thus I fear the comparison to other methods is not fair.
- Regarding the reproducibility, this approach has introduced many extra hyperparameters to be tuned, to name a few, the hyperparameters in Grad-Cam, $\operatorname{threshold}$ in Eq. 1, $1/8$ in Figure 3, $1/2$ in Eq. 4 and 6, the Mixup hyperparameter in Figure 3, $\xi$ in Eq. 7, and $\lambda$ in Eq. 8. Given that there are so many hyperparameters, I highly doubt the reproducibility of this approach.
- Furthermore, the robustness of the hyperparameters should be reported. Thus, not only accuracy, but also the variation.

#### 3. Experiments.

- The experiments seem extensive. But after a careful check, I found that there is no reason to compare with different SOTA results in different datasets (e.g., the comparison methods in PACS and Office-Home are different). Plus, the model selection strategy is also different for each dataset. There is no support or explanation in the paper to illustrate why this is done.
- It is unknown where the authors obtain the results for comparison methods for the main paper. Then, if you can use DomainBed, why not just use it as the main benchmark?

#### References

Overall, the references are good. But you can also cite more recent works from the following DG survey articles:

[1] Zhou et al. Domain generalzation in vision: a survey.

[2] Wang et al. Generalizating to unseen domains: a survey on domain generalization.

#### Minor comments

Typos:
- L26, "has attract".
- L137, "we propose develop".

---

> ### Author Response · Authors · 2022-08-02
> **Response to Reviewer Ywq6  (Part 3 -- Experiments)**
>
> > **Q1. The experiments seem extensive. But after a careful check, I found that there is no reason to compare with different SOTA results in different datasets (e.g., the comparison methods in PACS and Office-Home are different). Plus, the model selection strategy is also different for each dataset. There is no support or explanation in the paper to illustrate why this is done.**
>
> As stated in Section 4.2, to facilitate a fair and comprehensive comparison, we compare the proposed method to five types of state-of-the-art DG methods (almost cover all types of DG methods in the deep learning era). Please refer to Section 2 for the taxonomy of DG methods. For each type, we select 2 or 3 representative methods. Although PACS and Office-Home are two widely-used DG benchmarks, not all comparison methods have reported results on both datasets. As a consequence, we compare different SOTA results in different datasets, making sure that we could cover baseline methods as much as possible. In the meantime, these comparison methods follow the Train and Val splits established by several earlier works (cf. Section 4.1), and thus we strictly follow the mainstream settings (including their model selection strategies) to conduct our experiments. In our paper, we highlight the model selection strategy in the caption of each table so as to remind the readers how these comparison methods conduct their experiments.
>
> Given the above discussions, we argue that both the comparison methods and model selection strategy used in our main paper are reasonable and fair. Moreover, we have provided all these details in our original submission.
>
> >**Q2. It is unknown where the authors obtain the results for comparison methods for the main paper. Then, if you can use DomainBed, why not just use it as the main benchmark?**
>
> In our main paper, we follow MixStyle [1] (https://github.com/KaiyangZhou/mixstyle-release) to conduct experiments, including data preparation, model training, and selection. To facilitate a fair comparison, we select comparison methods that have open-source codes and make sure that the results for these comparison methods are obtained under the same evaluation protocol.
>
> We have also reported the results on DomainBed [2] benchmark (cf. Table 1 in the supplementary). However, as we all know, DomainBed is computationally expensive and requires about 4,142 models per DG algorithm, and thus we fail to finish all experiments before the main paper deadline. We are very willing to move the results based on DomainBed to the main paper in the revised version if it could help the readers to better understand our contributions.
>
> ***
>
> **Reference:**
>
> [1] Domain Generalization with MixStyle. In ICLR, 2021.
>
> [2] In Search of Lost Domain Generalization. In ICLR, 2021.

---

> > ### Comment · Reviewer_Ywq6 · 2022-08-03
> > **Results**
> >
> > I appreciate the author's response. But I think a fair comparison will be comparing with all common benchmarks, instead of specific choice on different datasets. Thus, it is better to put them all with the same baselines. I also appreciate the explanation on using mixstyle codebase.

---

> > > ### Author Response · Authors · 2022-08-07
> > > **Response to Reviewer Ywq6 (Results)**
> > >
> > > We understand the reviewer's concern and believe that the results on DomainBed (in the supplementary)  is capable of addressing this potential defect. On the other hand, we would like to raise the reviewer's attention that some of state-of-the-art methods only compare to small number of baseline methods. For example, EFDM [1] compares to three baseline methods and SagNet [2] compares five baseline methods. By contrast, in our main paper, we try to include almost all types of DG methods, resulting in the relatively different comparison methods on different datasets.
> > >
> > > ***
> > >
> > > **Reference:**
> > >
> > > [1] Exact Feature Distribution Matching for Arbitrary Style Transfer and Domain Generalization. In CVPR, 2022.
> > >
> > > [2] Reducing Domain Gap by Reducing Style Bias. In CVPR, 2021.

---

> ### Author Response · Authors · 2022-08-02
> **Response to Reviewer Ywq6  (Part 2 -- Overclaims)**
>
> > **Q1. In L36, how to evaluate that MiRe does not have the "data-dependent'' spurious correlations? I fear that MiRe is still a data-driven method and it heavily rely on the Mixup operation of the fore-background.**
>
> We have highlighted that spurious correlations can be alleviated but is difficult to be absolutely eliminated (footnote of the second page). In that sense, we cannot guarantee that MiRe does not have the "data-dependent'' spurious correlations in many real-world cases. To solve this issue, we introduce the concept of semantic topology that is robust to domain variations, implicitly reducing the bias towards data-dependent spurious correlations.
>
> > **Q2. In L42, authors claimed that previous methods require "some distribution of values for an attribute". I would assume MiRe does not. However, the adoption of Grad-Cam indeed assumes that the input data is image with clear fore and background information. Thus, it is an overclaim.**
>
> We agree that the adoption of Grad-Cam assumes that the input data is an image with clear fore and background information. In fact, the attribute here refers to the middle-level features which humans use to describe objects. We greatly appreciate this suggestion, and we have tune-downed our claim in the updated submission, as shown in blue lines in the revised manuscript.
>
> > **Q3. In L44, I fail to find the answers to the question: "how many factors of the latent factors".**
>
> Sorry for the confusion. This sentence is not a question. Here, we aim to highlight that in real-world cases, we cannot explicitly know the latent feature is composed of how many factors, and thus disentanglement-based methods cannot work well.
>
> > **Q4. In L132, why do existing data augmentation approaches lack diversity? Any support?**
>
> The scope of this statement is limited to existing augmentation-based DG methods [1-4]. As stated in the main paper, these methods typically randomly change the image style (e.g., color and texture) while maintaining the content unchanged. In other words, their augmentation strategies rely on manipulating the low-level feature statistics. By contrast, our CDM directly mixes the foreground and background of images from different domains. we have tune-downed our claim in the updated submission, as shown in blue lines in the revised manuscript.
>
> ***
>
> **Reference:**
>
> [1] Reducing domain gap by reducing style bias. In CVPR, 2021.
>
> [2] Addressing model vulnerability to distributional shifts over image transformation sets. In ICCV, 2019.
>
> [3] Exact feature distribution matching for arbitrary style transfer and domain generalization. In CVPR, 2022.
>
> [4] Domain generalization with mixstyle. In ICLR, 2021.

---

> > ### Comment · Reviewer_Ywq6 · 2022-08-03
> > **Response on overclaim**
> >
> > 1. I think you should avoid the usage of "data-dependent" since this method is definitely data-dependent. Echoing my previous response, this approach will fail if applied to non-image samples. Thus, it is data-dependent. Specifically, why do you think data-dependent is not good? I do not agree that previous data-dependent approaches are data-dependent is not bad.
> > 2. With regards to diversity, I think you should go beyond images and think more boldly: where does the diversity come from and how we can respond to them?

---

> > > ### Author Response · Authors · 2022-08-07
> > > **Response to Reviewer Ywq6 (Overclaim)**
> > >
> > > > **Q1. I think you should avoid the usage of "data-dependent" since this method is definitely data-dependent. Echoing my previous response, this approach will fail if applied to non-image samples. Thus, it is data-dependent. Specifically, why do you think data-dependent is not good? I do not agree that previous data-dependent approaches are data-dependent is not bad.**
> > >
> > > Thanks for your kind advice. In our context (L36 in the main paper), data-dependent means that the types of spurious correlations may be distinct across different datasets. More specifically, different datasets usually have different domain-specific factors, and the potential spurious correlations between these factors and the semantic label may be distinct across datasets. Thus, we call it ''data-dependent spurious correlations''. Here, this word might lead to some misunderstandings and we will remove it in the final version.
> > >
> > > **Q2. With regards to diversity, I think you should go beyond images and think more boldly: where does the diversity come from and how we can respond to them?**
> > >
> > > Thank you for this great question. Let us use sound separation as an example. Individual sounds are usually mixed with background noises and it is prohibitively difficult to directly disentangle them without prior knowledge of the source characteristics [1]. In this case, the idea of CDM (i.e., putting the target object in different backgrounds via recombination) can also applied to improve the diversity of training data and thus benefit the task of (unsupervised) sound separation.
> > >
> > >
> > >
> > > [1] Unsupervised Sound Separation Using Mixtures of Mixtures. In NeurIPS, 2020.

---

> ### Author Response · Authors · 2022-08-02
> **Response to Reviewer Ywq6  (Part 1-2 -- Methodology)**
>
> > **Q2. In figure 3, why only cropping 1/8 of the original image? There lacks motivation.**
>
> In experiments, we found that the generalization performance is insensitive to the ratio of cropping. We empirically verified this point by conducting experiments on PACS and Office-Home.
>
> | cropping value | PACS | Office-Home | Avg  |
> | -------------- | :--- | ----------- | :--- |
> | 1/4            | 83.9 | 65.8        | 74.9 |
> | 1/6            | 84.5 | 66.2        | 75.4 |
> | 1/8            | 84.8 | 66.2        | 75.5 |
> | 1/10           | 84.1 | 66.0        | 75.1 |
> | 1/12           | 84.3 | 66.1        | 75.2 |
> | 1/14           | 84.5 | 66.2        | 75.4 |
> | 1/16           | 84.0 | 66.1        | 75.1 |
>
> From the table, we can see that varying the cropping value in range [1/4, 1/16] just incurs a maximum variation of  0.6% in average classification accuracy. The justification is that images that serve as the backgrounds are first processed by Gaussian smoothing, making the them less informative compared to that of foreground images.
>
> > **Q3. Similarly, in Eq. 4, why adopting $cos(f,c_j)/2$? How did the term $1/2$ come up? This also goes to Eq. 6.**
>
> The term $1/2$ in Eq. 4 is used to average the local (the relations between each instance and all semantic anchors) and global (the relations among different semantic anchors) similarity. Similarly, term $1/2$ in Eq. 6 is used to average the cross-domain and cross-model similarity. Both local *vs.* global and cross-domain *vs.* cross-model are two orthogonal yet complementary parts. Here, we opt for a simple and commonly-used way to integrate these complementary parts.
>
> > **Q4. The second part of the method is too complicated and computationally expensive for multiple domains. Thus I fear the comparison to other methods is not fair.**
>
> In general, most cross-domain constraints in DG are first designed for two source domains and then extended to multiple domains. In our method, the second part introduces relation graphs to characterize the semantic topology in the embedding space. Here, we directly build the relation graphs on the top of prototype features and do not introduce any additional projection modules. Moreover, as the number of nodes of the relation graph equals to the number of classes in each dataset, the constructed graph is relatively small. The computational cost of edge connections (i.e., Eq. 4 and Eq. 6) is negligible due to the limited number of graph nodes. Thus, we believe the proposed ASTR is not computationally expensive. Also, considering the size and computational cost of the constructed relation graph, ASTR does not inferior to its competitors in terms of simplicity.
>
> > **Q5. Regarding the reproducibility, this approach has introduced many extra hyperparameters to be tuned, to name a few, the hyperparameters in Grad-Cam, threshold in Eq. 1, 1/8 in Figure 3, 1/2 in Eq. 4 and 6, the Mixup hyperparameter in Figure 3, $\xi$ in Eq. 7, and $\lambda$ in Eq. 8. Given that there are so many hyperparameters, I highly doubt the reproducibility of this approach.**
>
> **(i)** To the best of our knowledge, there is no hyper-parameter in Grad-Cam that should be tuned. **(ii)** 1/8 in Figure 3 and 1/2 in Eq. 4 and 6 are default designs and cannot be seen as hyper-parameters in our method. **(iii)** The mixup parameter is computed from the activation maps and cannot be seen as a hyper-parameter in our method. It is an adaptive parameter and do not require pre-definition. **(iv)** To this end, we think there are three hyperparameters that need to be discussed, i.e., threshold in Eq. 1, $\xi$ in Eq. 7 and $\lambda$ in Eq. 8. We will discuss the robustness of these hyperparameters in the following question.
>
> > **Q6. Furthermore, the robustness of the hyperparameters should be reported. Thus, not only accuracy, but also the variation.**
>
> We greatly appreciate this suggestion, and we have provided experimental results regarding the robustness of the hyperparameters in the revised supplementary.
>
> ***
>
> **Reference:**
>
> [1]  mixup: Beyond empirical risk minimization. In ICLR, 2018.
>
> [2] Domain Generalization with MixStyle. In ICLR, 2021.
>
> [3] Exact feature distribution matching for arbitrary style transfer and domain generalization. In CVPR, 2022.
>
> [4] Semi-supervised classification with graph convolutional networks. In ICLR, 2017.

---

> > ### Comment · Reviewer_Ywq6 · 2022-08-03
> > **Thank you for your clarification**
> >
> > 1. From the table of cropping 1/8 results, it seems 1/6 can also work well (only .1 difference). Therefore, 1/8 is a hyperparameter, right? But in your response to Q5, you said that 1/8 is not a hyperparameter, which seems in contradictory. Can you explain this?
> > 2. My question "why adopting $cos(\cdot, \cdot)$" distance is still not answered.
> > 3. Regarding the computational concern, why not show the time complexity and memory consumption, which will be more obvious.
> > 4. I appreciate the response to the hyperparameter robustness.

---

> > > ### Author Response · Authors · 2022-08-07
> > > **Response to Reviewer Ywq6**
> > >
> > > > **Q1. From the table of cropping 1/8 results, it seems 1/6 can also work well (only .1 difference). Therefore, 1/8 is a hyperparameter, right? But in your response to Q5, you said that 1/8 is not a hyperparameter, which seems in contradictory. Can you explain this?**
> > >
> > > Thank you for pointing out this problem. In experiments, we set the cropping ratio to a fixed value and **do not tune it on any datasets**. Thus, we said it is not a hyperparameter in our original design. In our response, following your suggestions, we evaluated the robustness of the proposed method to the variations of this parameter. To avoid the potential confusion or even contradiction, this parameter will be seen as a hyperparameter in our final version.
> > >
> > > > **Q2. My question "why adopting cos(⋅,⋅)" distance is still not answered.**
> > >
> > > We apologize for missing this question. Computing the similarity of two prototypes (under the presence of domain shifts) in the shared latent space based on cosine distance is a common method since cosine distance is domain-unrelated and insensitive to the feature dimension.
> > >
> > > > **Q3. Regarding the computational concern, why not show the time complexity and memory consumption, which will be more obvious.**
> > >
> > > Thanks for your kind advice. As suggested, we compare the proposed method to several state-of-the-art DG methods in terms of number of parameters (\#params.) and FLOPs. We report the comparison results on the PACS dataset (ResNet-18) as follows.
> > >
> > > | Method       | \#params. | FLOPs |
> > > | ------------ | :-------- | ----- |
> > > | SagNet [1]   | 22.7M     | 4.3G  |
> > > | MixStyle [2] | 11.2M     | 2.1G  |
> > > | EFDM [3]     | 11.2M     | 1.9G  |
> > > | Ours         | 12.2M     | 2.3G  |
> > >
> > > ***
> > >
> > > **Reference:**
> > >
> > > [1] Reducing Domain Gap by Reducing Style Bias. In CVPR, 2021.
> > >
> > > [2] Domain Generalization with MixStyle. In ICLR, 2021.
> > >
> > > [3] Exact Feature Distribution Matching for Arbitrary Style Transfer and Domain Generalization. In CVPR, 2022.

---

> > > > ### Comment · Reviewer_Ywq6 · 2022-08-08
> > > > **Thanks for your feedback**
> > > >
> > > > I appreciate your answers to other questions. But I think 1/8 is definitely a hyperparameter even if you do not tune it (then other people can, right?).
> > > >
> > > > There should be an ablation study in the future (not in this rebuttal given so limited time left) to validate metrics other than cosine.
> > > >
> > > > Appreciate your memory and flops comparison table.

---

> > > > > ### Author Response · Authors · 2022-08-08
> > > > > **Thanks**
> > > > >
> > > > > We thank Reviewer Ywq6 for the prompt response and engagement. Much appreciated. Please let us know if you have further comments or suggestions that have an influence on the final rating.

---

> ### Author Response · Authors · 2022-08-02
> **Response to Reviewer Ywq6  (Part 1-1 -- Methodology)**
>
> We thank the reviewer for the provided comments. After reading the comments seriously, we are afraid that the reviewer has probably misunderstood the contributions and the novelty of our paper. We will try our best to eliminate the misunderstandings via the following responses, and sincerely hope that the reviewer raises any further questions if he/she is still confused by our answers.
>
> > **Q1. Technically, the method is not novel. It is a combination of Grad-Cam, Mixup, and GCN. I admire the application of these existing techniques. But there is no further insight and motivation is not strong.**
>
> We would like to clarified that our contributions do not lie in the combination of  these existing techniques. We emphasize the  novelty and significance of our paper from the following two aspects:
>
> * The proposed CDM is novel in terms of its motivation and implementation. Mixing data in virtue of the complementary effect of between class and domain labels is novel and has never been explored in DG before as far as we know (also pointed out by **Reviewer Rsk7**). To achieve this goal, we introduce Grad-Cam to obtain the attentive regions of different classification losses and properly fuse them for depicting the complete foregrounds. Then, we mix the foreground and background of two images from different domains, which is significantly different from the classic mixup [1]. First, in contrast to [1] which requires a pre-defined balancing parameter, our CDM leverages the computed activation maps to guide the data fusion process in an adaptive manner. Second, CDM does not linearly combine the corresponding one-hot label of samples when mixing two samples. Instead, we still use the category label of the foreground object. The goal of this operation is to reduce the background bias, i.e., the potential spurious correlations between foreground and background regions. Finally, we have empirically verified that CDM is compatible with existing augmentation-based methods (e.g., MixStyle [2] and EFDM [3]) to further boost their performance (cf. Table 6 in main paper).
> * A long-standing and challenging problem for DG  (also for domain adaptation) is what should be transferred across domains. Prior works, such as adversarial training and statistic matching, focus on enforcing *one-vs-one* invariance, i.e., the representations of samples from the same semantic category should be invariant across domains. In this paper, we propose the concept of structural (*many-vs-many*) invariance on the top of semantic topology, i.e., the semantic relations between different categories should also be maintained across domains. This is exactly one of the motivations and the novelties of this paper (also pointed out by **Reviewer Rsk7 and GJ2F**) and has been clearly presented in our original submission. Technically, we instantiate the semantic topology as relation graphs and design the ASTR module to progressively refine the representations (node) and the topological relations (edge) of semantic anchors. The local feature aggregation and global cross-domain relational reasoning modules are proposed to perceive and maintain structural semantic relations. Here, spectral graph convolution [4] is introduced to perform graph representation learning. In a nutshell, instead of the simple application of GCN, our contributions lie in the instantiation and exploration of semantic topology via graphical structures.

---

> > ### Comment · Reviewer_Ywq6 · 2022-08-03
> > **Motivation of mixup is still not clear**
> >
> > Thanks for this detailed feedback! I still have the following questions:
> >
> > 1. The idea of "Mixing data in virtue of the complementary effect of between class and domain labels" is not explored before. But the motivation is still not clear since this method cannot be generic: it can only be applied to images, while other mixup-based approaches can be applied to all domains of data. Therefore, I question the applicability of this approach. If the inputs are time series or natural language, then this approach will fail.
> > 2. Indeed, when applying mixup to two samples, we often do not operate on the label space; instead, we use the loss mixup. This can be seen a trick.

---

> > > ### Author Response · Authors · 2022-08-07
> > > **Response to Reviewer Ywq6 (Motivation)**
> > >
> > > Thank you for the responsive reply. We hope these discussions could help the reviewer reconsider the score and would be very happy to answer any further questions. We answer your additional questions as follows.
> > >
> > > > **Q1. The idea of "Mixing data in virtue of the complementary effect of between class and domain labels" is not explored before. But the motivation is still not clear since this method cannot be generic: it can only be applied to images, while other mixup-based approaches can be applied to all domains of data. Therefore, I question the applicability of this approach. If the inputs are time series or natural language, then this approach will fail.**
> > >
> > > * We agree that CDM is originally designed for vision tasks, such as image classification (natural and medical images in the main paper) and semantic segmentation (urban scene understanding in the supplementary). In this regard, experiments in the main paper have extensively evaluate the effectiveness of CDM. Due to the significant difference between image and other modalities (such as time series and natural language), it is prohibitively difficult for us to explore CDM in broader ranges. In fact, we do not claim that our CDM is a variant of mixup and can be applicable to different modalities. Instead, our motivation is to solve the potential spurious correlations in **images** (object recognition tasks), which has been clearly stated and verified in our introduction and experiments.
> > >
> > > * On the other hand, the logic that 'the applicability of a DG approach is questionable as it cannot applied to different modalities' also goes for a number of state-of-the-art methods. For example, EFDM [1] proposes to match the empirical Cumulative Distribution Functions of image features. SagNet [2] disentangles style encodings from class categories to prevent style biased predictions and focus more on the contents of images. JiGen [3] solves the task of object recognition across domains by introducing self-supervised signals regarding how to solve a jigsaw puzzle on the same images. Similarly, our paper focuses on the task of **object recognition** across domains, which is an important yet challenging problem for the DG community.
> > >
> > > * At a last point, the motivation of mixing data for addressing domain generalized object recognition tasks is clear and has been demonstrated in our main paper. First, conventional DG methods strive to enforce domain- or class-wise invariance but are susceptible to include some misleading spurious correlations as the complex combinations of domain-specific and common factors lack in-depth exploration. Second, many recent efforts are devoted to disentangles common and domain-specific factors but face critical challenges in the real-world cases (cf. L34-52).
> > >
> > > Overall, the proposed CDM focuses on the task of object recognition across domains and can be applied to both natural and medical images.
> > >
> > > > **Q2. Indeed, when applying mixup to two samples, we often do not operate on the label space; instead, we use the loss mixup. This can be seen a trick.**
> > >
> > > Vanilla mixup aims to conduct data interpolation via convex combinations of pairs of examples and **their labels**. By contrast, the proposed CDM targets on generating diverse training samples by replacing the background of a certain image with a randomly cropped patch from other images but keeps its object label fixed.
> > >
> > > ***
> > >
> > > **Reference:**
> > >
> > > [1] Exact Feature Distribution Matching for Arbitrary Style Transfer and Domain Generalization. In CVPR, 2022.
> > >
> > > [2] Reducing Domain Gap by Reducing Style Bias. In CVPR, 2021.
> > >
> > > [3] Domain Generalization by Solving Jigsaw Puzzles. In CVPR, 2019.

---

### Official Review · Reviewer_GJ2F · 2022-06-29

**Rating:** 5
**Confidence:** 4
**Soundness:** 2 fair
**Presentation:** 2 fair
**Contribution:** 3 good

**Summary:**

This paper proposes Category-aware Data Mixing (CDM) to augment data on the data level and Adaptive Semantic Topology Refinement (ASTR) to maintain an invariant semantic topology of classes across different domains. The method (combination of CDM and ASTR) achieves excellent results on multiple DG datasets. Besides, experiments also show that CDM can bring performance gains to several other DG methods.

**Questions:**

Please see weaknesses for my concerns.

**Limitations:**

The authors didn't address the limitations and potential negative societal impact of their work.
- It is better to point out potential limitations, e.g., the proposed method (ASTR) can't work when the domain label is unavailable.
- I didn't see potential negative societal impact.

**Strengths And Weaknesses:**

Strengths (originality, quality, clarity and significance):
1. As far as I know, the originality is good.
2. The proposed method is technically sound. The experimental results compared with other DG methods is excellent.
3. This paper is generally easy to follow.
4. The idea of maintaining semantic topology of classes is novel.

Weaknesses:
1. The standard deviation (std) over 10 runs is missing. Considering most datasets are small, such as PACS and VLCS, it is more convincing to report the average accuracy with std for comparing model performances.
2. Some opinions in this paper are weak and unconvincing:
- Domain-wise invariance cannot guarantee generalizable representations. The authors argue that such invariance may be susceptible to including some misleading spurious correlations. This situation may occur in simulated data, where some semantically independent properties exist across all source domains. However, spurious correlation is difficult to hold simultaneously across all source domains in real-world cases, so it is reasonable to expect feature extractors to learn more semantic information through domain-invariant representation learning.
- The widely-adopted style-content-separation idea may fail to extract true semantic factors. The authors’ basis is that the activation map induced by domain classification does not focus on the background. This phenomenon is easy to understand because the style of foreground objects is also related to domain classification. However, I think that whether domain classification focuses on the background or not has no direct relationship with whether style-content-separation helps extract semantic information.
3. In Table 1 – Table 6, the authors should report DeepALL results that are implemented by themselves with the same training strategy as the proposed method. It is unfair to compare other methods with the extremely low DeepALL baseline, especially in Table 5. According to my experiments, it is easy for DeepALL to achieve an average accuracy above 82% with the ResNet-18 backbone on the PACS dataset. Experiments on DomainBed also report excellent performances of DeepALL (ERM).

---

> ### Author Response · Authors · 2022-08-02
> **Response to Reviewer GJ2F (Part 2)**
>
> > **Q3. In Table 1 – Table 6, the authors should report DeepALL results that are implemented by themselves with the same training strategy as the proposed method. It is unfair to compare other methods with the extremely low DeepALL baseline, especially in Table 5. According to my experiments, it is easy for DeepALL to achieve an average accuracy above 82% with the ResNet-18 backbone on the PACS dataset. Experiments on DomainBed also report excellent performances of DeepALL (ERM).**
>
> Thanks for your kind advice. According to our experiments, DeepAll which achieves an average accuracy above 82% with the ResNet-18 backbone on the PACS dataset is very likely to use the test domain for model selection (oracle). In our experiments, we strictly follow the train and val splits established by previous methods to conduct model selection, i.e., training-domain-validation setting. We summarize our DeepAll results as follows:
>
> **Table 1. Domain Generalization results on PACS benchmark.**
>
> | Art          | Cartoon      | Photo        | Sketch       | Avg  |
> | ------------ | :----------- | ------------ | :----------- | ---- |
> | 77.0$\pm$0.3 | 74.8$\pm$0.5 | 95.8$\pm$0.1 | 70.0$\pm$0.6 | 79.4 |
>
> **Table 2. Domain Generalization results on VLCS benchmark.**
>
> | VOC          | LabelMe      | Caltech      | Sun          | Avg  |
> | ------------ | :----------- | ------------ | :----------- | ---- |
> | 71.8$\pm$0.6 | 61.1$\pm$0.4 | 95.8$\pm$0.2 | 62.5$\pm$0.6 | 72.8 |
>
> **Table 3. Domain Generalization results on Office-Home benchmark.**
>
> | Art          | Clipart      | Product      | Real         | Avg  |
> | ------------ | :----------- | ------------ | :----------- | ---- |
> | 59.4$\pm$0.4 | 48.0$\pm$1.1 | 72.7$\pm$0.5 | 75.3$\pm$0.4 | 63.9 |
>
> **Table 4. Domain Generalization results on DomainNet benchmark.**
>
> | Clipart      | Infograph    | Quickdraw    | Painting     | Real         | Sketch       | Avg  |
> | ------------ | :----------- | ------------ | ------------ | ------------ | :----------- | ---- |
> | 61.8$\pm$0.6 | 20.2$\pm$0.7 | 45.0$\pm$1.3 | 14.3$\pm$1.0 | 57.3$\pm$0.7 | 44.8$\pm$0.6 | 40.6 |
>
> The above results have been added to the revised manuscript.

---

> > ### Comment · Reviewer_GJ2F · 2022-08-08
> > **Response to Authors**
> >
> > Thanks for the authors' responses. The authors have addressed my main concern regarding Part 1. However, I am still doubtful about the results of ERM on PACS and VLCS datasets in Part2. I trained the model for 50 epochs and used the last-step checkpoint (same as [1]). For the ResNet-18 backbone, test domain accuracies are 82.7% on PACS and 74.5% on VLCS, respectively, without any complex data augmentation. For the ResNet-50 backbone, my result on the PACS dataset is consistent with the reported result in [2] with training-domain validation set as model selection.
> >
> > Do you keep the training parameters of DeepAll the same as MiRe's? I believe MiRe's hyperparameters are fully tuned, but I am not sure about your DeepAll training strategy.
> >
> > [1] K. Zhou, Y. Yang, Y. Qiao, and T. Xiang, “Domain Generalization with MixStyle,” in ICLR, 2021.
> >
> > [2] I. Gulrajani and D. Lopez-Paz, “In Search of Lost Domain Generalization,” in ICLR, 2021.

---

> > > ### Author Response · Authors · 2022-08-08
> > > **Response to Reviewer GJ2F**
> > >
> > > Thank you for your time reviewing our paper and for your valuable discussions. We answer your additional questions as follows.
> > >
> > > > **Q1. The results of ERM on PACS and VLCS datasets in Part2.**
> > >
> > > According to the the reported results of a number of previous DG works and the careful checking of our source codes many times, we believe that the ERM results in our paper are reasonable. We do appreciate it if the reviewer could double-check his/her codes, e.g., whether the parts of MixStyle [1] are commented out (L197, L200, and L203 in https://github.com/KaiyangZhou/Dassl.pytorch/blob/master/dassl/modeling/backbone/resnet.py). Noting that the original training-validation split (https://drive.google.com/drive/folders/0B6x7gtvErXgfUU1WcGY5SzdwZVk?resourcekey=0-2fvpQY_QSyJf2uIECzqPuQ) provided by [2] is used. Also, we used the last-step checkpoint to perform an evaluation on the target domain and obtained the following results, which have a negligible difference compared to the training-domain validation setting (i.e., select the model that exhibits the best performance in the validation dataset).
> > >
> > > | Method              | Art  | Cartoon | Photo | Sketch | Avg  |
> > > | ------------------- | :--- | ------- | :---- | ------ | ---- |
> > > | DeepAll (last step) | 78.9 | 75.6    | 96.2  | 67.0   | 79.4 |
> > >
> > > We provided the trained models and corresponding training logs in this anonymous link (https://drive.google.com/file/d/188MlrDgywunE8nDtD4o0BaxMlIGtPJ01/view?usp=sharing).
> > >
> > > On the PACS dataset, we used the source codes of MixStyle and trained the model for 50 epochs. Then, we used the last-step checkpoint and obtained the following results.
> > >
> > > | Method   | Art  | Cartoon | Photo | Sketch | Avg  |
> > > | -------- | :--- | ------- | :---- | ------ | ---- |
> > > | MixStyle | 82.1 | 78.6    | 96.9  | 73.3   | 82.7 |
> > >
> > > > **Q2. Do you keep the training parameters of DeepAll the same as MiRe's? I believe MiRe's hyperparameters are fully tuned, but I am not sure about your DeepAll training strategy.**
> > >
> > > We are sure that the training parameters of DeepAll are the same as MiRe's after a careful check.
> > >
> > > ***
> > >
> > > [1] Domain Generalization with MixStyle. In ICLR, 2021.
> > >
> > > [2] Deeper, broader and artier domain generalization. In ICCV, 2017.

---

> > > ### Author Response · Authors · 2022-08-09
> > > **Final thought**
> > >
> > > Dear Reviewer GJ2F,
> > >
> > > We are wondering if you could give some comments and final thoughts about our previous discussion. Please let us know if you have further follow-up discussions. We would be immensely grateful if you could raise the rating to reflect the contributions of our paper.
> > >
> > > Thank you very much.
> > >
> > > Best regards,
> > >
> > > Authors

---

> ### Author Response · Authors · 2022-08-02
> **Response to Reviewer GJ2F (Part 1)**
>
> We thank the reviewer for helping improve our paper and appreciate that they recognized the novelty and value of our work. We address the concerns as follows and will revise the manuscript accordingly. We sincerely hope that the reviewer raises any further questions if he/she is still confused by our answers.
>
> > **Q1. The standard deviation (std) over 10 runs is missing. Considering most datasets are small, such as PACS and VLCS, it is more convincing to report the average accuracy with std for comparing model performances.**
>
> Thanks for your kind advice. We have included results of std (over 10 runs) in Table 1-4 in the revised manuscript. We summarize the results as follows:
>
> **Table 1. Domain Generalization results on PACS benchmark.**
>
> | Art          | Cartoon      | Photo        | Sketch       | Avg  |
> | ------------ | :----------- | ------------ | :----------- | ---- |
> | 84.6$\pm$0.5 | 79.5$\pm$0.4 | 96.8$\pm$0.2 | 78.4$\pm$1.0 | 84.8 |
>
> **Table 2. Domain Generalization results on VLCS benchmark.**
>
> | VOC          | LabelMe      | Caltech      | Sun          | Avg  |
> | ------------ | :----------- | ------------ | :----------- | ---- |
> | 70.3$\pm$0.3 | 63.6$\pm$0.7 | 96.2$\pm$0.4 | 69.4$\pm$0.7 | 74.9 |
>
> **Table 3. Domain Generalization results on Office-Home benchmark.**
>
> | Art          | Clipart      | Product      | Real         | Avg  |
> | ------------ | :----------- | ------------ | :----------- | ---- |
> | 60.2$\pm$0.8 | 53.2$\pm$0.9 | 75.1$\pm$0.6 | 76.4$\pm$0.6 | 66.2 |
>
> **Table 4. Domain Generalization results on DomainNet benchmark.**
>
> | Clipart      | Infograph    | Quickdraw    | Painting     | Real         | Sketch       | Avg  |
> | ------------ | :----------- | ------------ | ------------ | ------------ | :----------- | ---- |
> | 64.7$\pm$1.0 | 27.8$\pm$1.2 | 53.1$\pm$0.8 | 16.4$\pm$1.5 | 64.1$\pm$0.6 | 52.3$\pm$0.9 | 46.4 |
>
> From the table, we can see that the proposed method is insensitive to the random seed on most benchmark datasets, revealing the efficacy and robustness of our data mixing and structural relation modeling modules.
>
> > **Q2. Some opinions in this paper are weak and unconvincing.**
>
> > Q2.1: Domain-wise invariance cannot guarantee generalizable representations. The authors argue that such invariance may be susceptible to including some misleading spurious correlations. This situation may occur in simulated data, where some semantically independent properties exist across all source domains. However, spurious correlation is difficult to hold simultaneously across all source domains in real-world cases, so it is reasonable to expect feature extractors to learn more semantic information through domain-invariant representation learning.
>
> We greatly appreciate this suggestion, and we have tune-downed our claim in the updated manuscript. Our key insight is that domain-wise and category-wise invariance are pairwise (one-vs-one) alignment strategies, which cannot explore the complex many-vs-many interactions, i.e., the relations of different semantic categories. In essence, the classical one-vs-one alignment (such as adversarial training) can be seen as the simplest case of relational reasoning. Most conventional DG methods assume that perfect alignment equals to precise knowledge transfer, while the many-vs-many relations between different entities are ignored. Moreover, alignment-based approaches naturally neglect the intra-domain relations as no entities can be explicitly aligned within each domain. In this regard, our work gives a hint to bridge the gap between alignment-based and relational reasoning based DG by jointly modeling the intra-domain and inter-domain topological relations.
>
> > Q2.2: The widely-adopted style-content-separation idea may fail to extract true semantic factors. The authors’ basis is that the activation map induced by domain classification does not focus on the background. This phenomenon is easy to understand because the style of foreground objects is also related to domain classification. However, I think that whether domain classification focuses on the background or not has no direct relationship with whether style-content-separation helps extract semantic information.
>
> Apologies for the misunderstanding caused by this sentence and we have modified the argument in question. We agree that whether domain classification focuses on the background or not has no direct relationship with whether style-content-separation helps extract semantic information. Here, we aim to highlight that the activation map induced by domain classification does not focus on the background.

---

### Official Review · Reviewer_Rsk7 · 2022-07-03

**Rating:** 8
**Confidence:** 5
**Soundness:** 3 good
**Presentation:** 3 good
**Contribution:** 3 good

**Summary:**

The paper proposes Mix and Reason, a new domain generalization approach consisting of two components. The first component, category-aware data mixing, basically performs data augmentation by replacing the background of an image with a random patch of another image, aided by binary masks produced by Grad-CAM. The second component, adaptive semantic topology refinement, is built using a graph neural network, which aims to learn semantic features through interactions among source domains. The effectiveness of the approach is demonstrated on a number of commonly-used domain generalization benchmarks.

**Questions:**

- Would it be possible to show some visualization on what the "semantic topology" looks like or what was learned by the ASTR module?
- In Table 7, only ASTR is evaluated. Does CDM also work for the medical datasets?
- Regarding the use of Grad-CAM, does it always give accurate masks? Are there any failure cases? And what happens if Grad-CAM doesn't produce accurate masks?

-------- Post-rebuttal updates --------

The authors' responses have resolved all the reviewer's concerns.

As mentioned in the 1st round review, the reviewer appreciated the contributions and believed the proposed approach and insights will be useful to the community (the reviewer's stance remains the same). The added materials during the rebuttal have significantly strengthened the motivation of the paper and made the story more convincing.

The reviewer has also read other reviewers' comments as well as the authors' responses, and found no critical issues that could lead to rejection.

Overall, the reviewer likes the contributions of the paper and believes they would be of interest to the domain generalization community. Therefore, the reviewer strongly recommends acceptance and stands firmly on this side.

**Limitations:**

Currently there is no discussion about the limitations of the approach. Perhaps there are some limitations related to the use of Grad-CAM (see the Questions section).

**Strengths And Weaknesses:**

**Originality**: From the technical point of view, the proposed ideas, including the data mixing strategy and the graph neural network-based module, are novel. It's interesting that Grad-CAM can be used in this way for data augmentation.

**Quality**: Overall the quality is good. The main experiments are comprehensive and solid for justifying the technical contributions. The ablation studies are also sufficient. It's worth mentioning that the datasets cover a wide range of tasks including generic object recognition and medical image analysis. However, the paper misses more in-depth analysis on "how" the graph neural network-based ASTR module improves the performance. The quantitative results are sufficient to justify the effectiveness but a more scientific analysis is needed to help readers understand how the "topology" is built by the neural network and in what way it helps learn structural invariance. In other words, the concept of semantic topology is new and sounds "big" but there is no concrete studies on how this concept works. (Maybe use a theory or visualize what was learned by the model?)

**Clarity**: The paper is well-written.

**Significance**: The paper provides new insights on how data augmentation can be combined with graph neural networks to improve domain generalization. Moreover, the results demonstrate that each of the two components alone could be useful to the community: the data mixing strategy works with some existing methods like MixStyle and EFDM, while the graph neural network can be used for medical image analysis.

---

> ### Author Response · Authors · 2022-08-02
> **Response to Reviewer Rsk7**
>
> Thank you for your insightful review and for recognizing our novel contributions. Below please find our point-by-point response to your comments.
>
> > **Q1. Would it be possible to show some visualization on what the "semantic topology" looks like or what was learned by the ASTR module?**
>
> We greatly appreciate this suggestion, and we have added some visualization results to the revised supplementary. As shown in Fig. 2 of the supplementary, we visualize the relations (edge weights) of semantic anchors (nodes) on the PACS dataset. From the figure, we can observe that those semantically similar anchors will be assigned with larger weights, while those dissimilar anchors will be assigned with small weights. More importantly, such relations will be generalizable across domains, fitting the human intelligence that humans are talented at comparing and reasoning when learning new concepts.
>
> > **Q2. In Table 7, only ASTR is evaluated. Does CDM also work for the medical datasets?**
>
> CDM requires that the images should be comprised of the foreground objects and the background. In that sense, not all medical datasets satisfy this requirement as some medical images do not have explicit lesion regions. For example, in the task of tuberculosis diagnosis, attributes like atelectasis and pulmonary cavitation cannot be characterized by class and domain labels. By contrast, tasks like benign-malignant classification of pulmonary nodules in chest CT images that have explicit lesion regions could be used to evaluate our CDM. We will try to include more medical datasets in the final version.
>
> > **Q3. Regarding the use of Grad-CAM, does it always give accurate masks? Are there any failure cases? And what happens if Grad-CAM doesn't produce accurate masks?**
>
> Grad-CAM aims to provide the attentive regions of domain and class classification losses. Here, we leverage their complementary effects to depict the complete foreground regions. However, we cannot guarantee that Grad-CAM will always give accurate masks. As the proposed CDM explicitly mixes the foreground and background regions of different images, the potential inaccurate masks could increase the diversity of the generated data. The ablation of CDM (cf. Table 5) and its combination with prior augmentation-based works (cf. Table 6) provide empirical support, verifying that CDM is robust and scalable to different domain generalization scenarios.

---

### Meta-Review · Area_Chair_X6XJ · 2022-08-28

**Recommendation:** Accept
**Confidence:** Less certain

**Metareview:**

Some of the reviewers had concerns about novelty, and one of the reviewers was worried about the care taken in training a baseline.  however, another reviewer has a strong positive opinion of the work; and I believe the authors have done a good job in rebuttal making an effort to address the concerns about baselines.  I am recommending acceptance; but I expect the authors to release code to allow further scrutiny w.r.t. baselines.

**Award:**

No

---

### Decision · Program_Chairs · 2022-09-14

Accept